# Intraarticular Implantation of Autologous Chondrocytes Placed on Collagen or Polyethersulfone Scaffolds: An Experimental Study in Rabbits

**DOI:** 10.3390/polym15102360

**Published:** 2023-05-18

**Authors:** Maciej Płończak, Monika Wasyłeczko, Tomasz Jakutowicz, Andrzej Chwojnowski, Jarosław Czubak

**Affiliations:** 1Mazovia Regional Hospital John Paul II, 08-110 Siedlce, Poland; 2Nałęcz Institute of Biocybernetic and Biomedical Engineering, Polish Academy of Sciences, 02-109 Warsaw, Poland; mwasyleczko@ibib.waw.pl (M.W.);; 3Department of Neurosurgery and Children Traumatology, Medical University of Warsaw, 02-091 Warsaw, Poland; jakutowicztom@gmail.com; 4Department of Orthopedics, Pediatric Orthopedics and Traumatology, Centre of Postgraduate Medical Education, Gruca Orthopaedic and Trauma Teaching Hospital, 05-402 Otwock, Poland

**Keywords:** autologous chondrocyte implantation, cartilage regeneration, cartilage tissue engineering, collagen scaffold, osteochondral defects, rabbit, polyethersulfone scaffold, regenerative medicine

## Abstract

Hyaline cartilage has very limited repair capability and cannot be rebuilt predictably using conventional treatments. This study presents Autologous Chondrocyte Implantation (ACI) on two different scaffolds for the treatment of lesions in hyaline cartilage in rabbits. The first one is a commercially available scaffold (Chondro–Gide) made of collagen type I/III and the second one is a polyethersulfone (PES) synthetic membrane, manufactured by phase inversion. The revolutionary idea in the present study is the fact that we used PES membranes, which have unique features and benefits that are desirable for the 3D cultivation of chondrocytes. Sixty-four White New Zealand rabbits were used in this research. Defects penetrating into the subchondral bone were filled with or without the placement of chondrocytes on collagen or PES membranes after two weeks of culture. The expression of the gene encoding type II procollagen, a molecular marker of chondrocytes, was evaluated. Elemental analysis was performed to estimate the weight of tissue grown on the PES membrane. The reparative tissue was analyzed macroscopically and histologically after surgery at 12, 25, and 52 weeks. RT-PCR analysis of the mRNA isolated from cells detached from the polysulphonic membrane revealed the expression of type II procollagen. The elementary analysis of polysulphonic membrane slices after 2 weeks of culture with chondrocytes revealed a concentration of 0.23 mg of tissue on one part of the membrane. Macroscopic and microscopic evaluation indicated that the quality of regenerated tissue was similar after the transplantation of cells placed on polysulphonic or collagen membranes. The established method for the culture and transplantation of chondrocytes placed on polysulphonic membranes resulted in the growth of the regenerated tissue, revealing the morphology of hyaline-like cartilage to be of similar quality to collagen membranes.

## 1. Introduction

Articular cartilage (AC) is the tissue that covers the long bones ends. It provides sufficient structural stability to transfer heavy loads between bones. It is strong, frictionless for load-bearing surfaces, and protects the subchondral bone. Unfortunately, it can be distorted, leading to adverse complications [1,2,3]. AC cannot regenerate because of the absence of nerves and blood vessels, and the poor mitogenic force of chondrocytes. Left untreated, damage to cartilage causes pain, stiffness, movement limitation, and disappearance of joint function. This can lead to diseases, including osteoarthritis, or even disability [1,2,4,5,6,7]. Available treatments do not do much to rebuild the quality and efficiency of the joint surface. The commonly used treatment methods of AC are microfracture (MF) and cell-based techniques, such as autologous chondrocyte implantation (ACI) or mosaicplasty [4,8,9,10]. The limited ability of AC to reclaim has received clinical awareness and interest in recent times. Many trials, like implantation of the periosteum, cells, or osteochondral fragments, have been performed in pre-clinical (animals model) or even in clinical attempts [11,12]. The MF technique provides infiltration of the mesenchymal stem cells from the bone marrow (subchondral bone perforation), whereby a blood clot fills in the damage. This provides a suitable environment for AC regeneration [13,14]. In the ACI method, the isolated chondrocytes from the host are placed on a scaffold and cultured in an incubator. After some weeks, the obtained bio-implant is transplanted. Achieving hyaline or hyaline-like repair by ACI for damaged AC is very promising. Research evidence shows that the transplantation of chondrocytes with membranes (scaffolds or matrices) obtained from natural or synthetic biomaterials can improve the quality of AC [4,9,10,15,16,17,18]. Unfortunately, in the MF and ACI techniques, the regeneration is composed mainly of fibrocartilage. It has less favorable biomechanical properties than hyaline cartilage, and can be exposed to further damage. Furthermore, experimental evidence shows better results with the ACI than with the MF method [10,19,20]. Thus, researchers and doctors are still searching for a more promising method of AC regeneration, among which the most promising one is to use cells with biomaterials (bioimplants) [1,4,8,9,16,17,18,21,22].

The membranes (scaffolds) must be spatial, with interconnected pore structures. They should be biocompatible, and possess sufficient stiffness, mechanical stability, and shape properties to withstand stresses during culture and after implantation [16,23,24,25,26,27]. Scaffold materials should be biocompatible, biodegrade to non-harmful components in the organism, and be resistant to body conditions such as pH and temperature. Therefore, it is necessary to choose appropriate biomaterials—synthetic or natural polymers, or their combination (hybrid materials) [9,16,25,26,28,29,30,31,32]. Natural polymers, such as collagen, hyaluronic acid (HA), chondroitin sulfate (CS), chitosan (CH), and fibrin [9,33,34,35,36,37], are distinguished by their high biocompatibility and bioactivity. Natural materials have features similar to those of human tissues, so they exist naturally in the human organism. Due to their origin, they stimulate cell adhesion and ECM production. Despite their many advantages, they have significant disadvantages. In the aqueous environment, they quickly lose mechanical properties (hydrolysis). The scaffold loses suitable properties for cell culture (supporting cells). In addition, methods for obtaining scaffolds from these materials are limited due to their lack of resistance to process parameters, like high temperature or pressure [9,16,24,25,26,27,28,33]. Synthetic polymers: polycaprolactone (PCL), polyurethane, polylactic acid (PLA), and polyethersulfone (PES) [21,31,38,39,40,41,42,43] are more diverse and promising. Compared to natural materials, they can be used to produce a variety of membrane structures using different methods. They have appropriate mechanical, physical, and chemical properties. Many of them degrade to non-toxic components in the body. In addition, the mechanical properties and degradation time of the scaffolds can be controlled through appropriate combinations of these polymers—copolymers or blends [16,27,28,29,33,44,45,46,47]. Hybrid scaffolds should also be briefly introduced. They combine the advantages of synthetic and natural biomaterials. This makes it possible to obtain scaffolds with appropriate mechanical properties, bio-functionality, and the ability to regulate degradation [16,25,26,30].

The purpose of the study was to evaluate the effect of autologous chondrocyte transplantation, placed on polyethersulfone (PES) or commercial, collagen Chondro-Guide scaffolds, in the treatment of lesions in hyaline cartilage in rabbits. An effective technique of isolation and culture of chondrocytes was established. Rabbits were divided into five groups. Full-thickness lesions were produced, and the scaffolds were used with or without autologous chondrocytes. As a control, non-filled cavities were used. The macroscopic and microscopic appearance of the tissue in the damaged region of the AC surface as a result of regeneration was evaluated. In the end, the chondrogenic potential of the cells cultured in vitro on a non-absorbable PES membrane was compared with that of cells cultured in vitro on an absorbable collagen scaffold. The regenerated tissue was observed over time, up to 52 weeks. The novelty of this study lies in the use of the synthetic PES membrane, which was used here for the first time on an animal model (rabbits). In addition, it is compared with a commercial membrane made of collagen. Studies show that the PES membrane was able to achieve its task, and can be used for further research.

## 2. Materials

### 2.1. Membranes

The semipermeable, porous membrane was made of a polyethersulfone (PES) (Figure 1). Polymer is biocompatible, and is used in tissue engineering [38,42,48].

The PES scaffold was designed and manufactured in cooperation with chemists from the Institute of Biocybernetics and Biomedical Engineering of the Polish Academy of Sciences in Warsaw. The choice of this material was determined by its favorable biomechanical properties, like chemical resistance and the possibility of forming a membrane using easy methods. It was obtained by the wet inversion phase technique according to the previous work [38,49,50]. Figure 2 shows the cross-section and the top and bottom layers of the PES and Chondro-Gide scaffolds. The top layer of the PES scaffold is perforated, while the bottom layer is smooth and compacted. This is necessary for scaffolds, as it allows cells to enter the membrane and be retained inside.

A Chondro-Gide membrane was used to comparatively evaluate the chondrogenic potential of the PES membrane (Figure 2). A commercial scaffold manufactured by the Swiss company Geistlich Pharma AG was employed. According to the information provided by the company, it is a collagen membrane made of type I and III collagen of porcine origin. It has a two-layer structure, comprised of a compact layer and a porous layer. The compact layer has a smooth surface visible on one side of the membrane, and the spongy layer is characterized by its rough surface. Due to the materials used, the Chondro-Gide is absorbable. The arrangement of collagen fibers increases the membrane’s resistance to tearing. The proteolytic enzyme collagenase is responsible for membrane resorption, and the resulting breakdown products are denatured at 37 °C. The resulting oligopeptides are broken down into individual amino acids. Membrane manufacturers have also described its immunogenic properties as being very low [51].

### 2.2. Rabbits

The right and left knees of 64 White New Zealand rabbits, weighing 2–3.5 kg and aged 4 months, were used for this study. They were kept in an animal house on the premises of the Miroslaw Mossakowski Institute of Experimental and Clinical Medicine of the Polish Academy of Sciences in Warsaw, under standard environmental conditions: air humidity 50 ± 10%, temperature 24 °C ± 2 °C. The animals were kept in separate cages, where they were able to move freely, both before and after treatment.

In all of the knee joints, grade IV defects on the articular surface were produced (Figure 3).

Operational joints were divided into five groups:Full-thickness defect with implanted chondrocytes placed on a collagen membrane: 28 knees.Full-thickness defect and chondrocytes implanted on a PES membrane: 30 knees.Full-thickness lesion and implantation of a collagen membrane without cells: 25 knees.Full-thickness defect and implantation of a PES membrane without cells: 26 knees.Full-thickness defect without any implant, allowing cells from the bone marrow to infiltrate the regenerated tissue: 13 knees.

The choice of research material was justified by the following facts: the morphological and functional similarity of rabbit and human articular cartilage; the relatively low cost of purchase and culture; ease of performing general anesthesia; the ubiquity of the use of rabbits for experimental studies on articular cartilage; and the global literature, providing the possibility of comparing our results with the work of other authors.

This research received the approval of the First Warsaw Ethical Commission for Experiments on Animals of the M. Nencki Institute of Experimental Biology of the Polish Academy of Sciences in Warsaw, through Opinion No. 349/2004.

## 3. Methods

### 3.1. Chondrocyte Isolation and Culture Techniques

For Group I and II rabbits, articular cartilage slices were taken from the non-weight-bearing area of the articular surface. Cartilage fragments were collected from the marginal region of the lateral and medial condyles of the rabbits’ femurs. They were transported to the laboratory in sterile tubes containing about 1.5 mL of saline solution (0.9% NaCl), where cell isolation and culture were carried out. The cell isolation process was initiated less than two hours after the time at which the tissue was collected. They were cut into slices approximately 1 mm thick using a surgical blade (No. 12) under the sterile conditions of a laminar chamber. The cartilage mass was then washed several times with saline solution and placed in a sterile tube containing 0.25% of collagenase type II solution with culture medium (RPMI with DNAase at a concentration of 7.2 g/L (17.6 units/g), 10% FBS serum, and 1.5% 100 × diluted antibiotics (Streptomycin and Penicillin)). Then, the sample was shaken for 12 h in the incubator (37 °C, 5% CO_2_). Subsequently, the obtained samples were centrifuged at 5 °C and 1000 rpm for 5 min. The supernatant was discarded, and the obtained pellets were suspended in 2 mL medium. Before cell counting with the Bürker chamber, the cells were stained with a 0.5% solution of trypan blue. Then, the cells were placed on a PES or collagen membrane with a diameter of 5 mm in a six-well cell culture plate. Supplementary medium was added to each well. Cells were incubated at 37 °C and 5% CO_2_. Implantation was performed 14 days after the biopsy.

### 3.2. Identification of Procollagen Type II

Total mRNA was isolated from the cultured cells according to Chomczyński’s method [53]. The obtained mRNA was reverse transcribed (RT-PCR) to obtain cDNA. Then, the cDNA was amplified by PCR, and the product was identified by electrophoresis. The expression of the gene encoding procollagen type II, which is a molecular marker of chondrocytes, was evaluated. Due to the presence of type I and III collagen in the collagen membrane, the test was performed using the PES membrane. The expression of the gene encoding procollagen type II, a molecular marker of chondrocytes, was evaluated only in the second group of rabbits, with implanted chondrocytes placed on a PES membrane. For collagen II, the PCR conditions differed with respect to the primer attachment temperature, which was 61 °C. All PCR reactions were carried out using a Perkin Elmer 9600 machine. The reaction products were developed by electrophoresis on a 1.5% agarose gel, and the resulting product was stained with ethidium bromide.

The primer sequences used for DNA amplification by PCR were as follows:Collagen I (5′ starter—5′-CCAGATTGAGACCCTCCTCA-3′, 3′starter—5′-ATGCAATGCTGTTCTTGCAG-3′)Collagen II (5′ starter—5′-GGGGTCCTTTAGGTCCTACG-3′, 3′starter—5′-AGTCGCTGGTGCTGCTGAC-3′)

### 3.3. Elemental Analysis

To estimate the mass of the tissue grown on the PES membrane after culture, elemental analysis was performed at the Department of Analytical Chemistry, Faculty of Chemistry, Warsaw University of Technology. The tissue mass of the 5-mm-diameter PES membrane slices were determined on the basis of nitrogen (N) content. The test was performed before (reference samples) and after chondrocyte culture (2 weeks). The post-culture samples were fixed with 2.5% glutaraldehyde and dried. Protein was assessed by multiplying the determined nitrogen content by a protein conversion factor of 6.25. Eight analyses were performed—four for the reference samples and four for scaffolds after culture. Due to the composition of the Chondro-Gide membrane (type I, III collagen), these tests were performed only for the PES scaffold.

### 3.4. SEM Observation

The PES samples were immersed in ethanol for about 15 min. They were then broken into pieces in liquid nitrogen. The samples prepared in this way were dried and coated with a 7 nm layer of gold using a sputtering machine.

### 3.5. Implantation of Grafts

The surgical procedures were performed under aseptic conditions in the operating room. Subsequent surgeries were performed on the rabbits in Groups I, II, III, IV, and V under combined general anesthesia, using 30 mg/kg ketamine hydrochloride and 2 mg/kg xylazine administered intramuscularly. The rabbits were placed in a supine position, and the surgery was performed on both knees. After shaving and sterile prepping of the lower limbs, an anteromedial parapatellar arthrotomy was performed in the knee. Each patella was dislocated laterally, and a cylindrical osteochondral defect penetrating the subchondral bone was made on the patellar groove of the femur. The defect was 5 mm in diameter and 4 mm in depth. In Group I, II, III, and IV rabbits, the defect was filled with PES or collagen membrane, which was rolled up and attached by press fitting. In Group V rabbits (control group), the defect was left empty. In Group I and II rabbits, autologous chondrocytes were implanted on both kinds of membranes into the defect. In Group III and IV rabbits, membranes without cells were used. The knee wound was closed in layers with 4–0 vicryl sutures. After the operation, all animals were allowed to walk freely in cages without splinting. The rabbits were terminated with an overdose of phenobarbital sodium salt at 12, 25, and 52 weeks after the operation.

### 3.6. Macroscopic Evaluation

The filling level of the defect relative to the surrounding healthy tissue, the degree of integration, and the regularity of the surface were evaluated macroscopically. For this purpose, the International Cartilage Repair Society (ICRS) cartilage repair assessment scale was used (Figure 4) [54].

### 3.7. Microscopic Analysis

Decalcified material was embedded in Paraplast PLUS paraffin (Sigma Aldrich, Poznań, Poland). The blocks were cut in the frontal plane onto slides using a microtome, to a thickness of 4 µm. Paraffin sections were stained by the routine hematoxylin–eosin method. Samples from several areas of the regeneration were selected for microscopic evaluation. The reparative tissue was analyzed histologically. The cell morphology, regularity of surface, integrity of the structure, thickness, integration with surrounding cartilage, cellularity, and necrosis were studied by microscopic analysis on the basis of the O’Driscoll scale (Table 1) [55,56].

### 3.8. Statistical Analysis

For macroscopic and microscopic data analysis, the arithmetical mean, standard deviation (SD), and median value were evaluated. The Mann–Whitney test was used to compare the scores of the experimental groups. A significance level of 0.05 was used for the statistical analysis.

## 4. Results

### 4.1. The Number of Cells

Cells obtained by enzymatic isolation were counted in a Bürker chamber. Approximately 1–1.5 × 10^5^ viable cells were obtained after each isolation. An equal number of cells were transferred to scaffolds in six-well plates.

After 14 days of culture, images were taken using an inverted microscope. Figure 5 shows cells that were attached to the edge of the membrane.

The presence of cells on the PES membranes was confirmed via SEM microscopy (Figure 6). Cells with an intercellular matrix are noticeable on the PES scaffold, as indicated by the red arrows.

### 4.2. RT-PCR for mRNA of Type II Collagen

Because of the material composing Chondro-Gide (collagen type I and III), the RT-PCR test was only performed for the PES scaffolds. The test was carried out after 2 weeks of cultivation. The analysis of the mRNA isolated from the cells from the bottom of the culture plate and the PES membrane revealed thick bands corresponding to 145 base pairs (bp). There was no band at 352 bp. These data indicate the expression of type II procollagen without exon 2 (type IIB collagen), which is characteristic of cartilage. The bands at 400 bp appearing in the RT-PCR for mRNA, which was extracted from the cells detached from the culture flask and the PES membrane, indicated the expression of type I procollagen. However, the thickness of the bands was greater for cells detached from the bottom of the culture flask (Figure 7).

### 4.3. Elementary Analysis

The elementary analysis was carried out only for the PES scaffolds. The problem here was the same as in the case of RT-PCR analysis. Collagen types I/III contain nitrogen, so the results would not be correct. The PES membrane slices after 2 weeks of culture with chondrocytes revealed a concentration of about 0.23 ± 0.035 mg of tissue on one part of the membrane (Table 2).

### 4.4. Macroscopic Evaluation

After 12, 28, and 54 weeks, most of the operated knee joints had normal contouring and preserved a full range of motion. According to the ICRS scale (Figure 4), a diagram was made (Figure 8). Each group of rabbits was considered in turn, and they were divided into subgroups on the basis of the number of points obtained. Data from the macroscopic evaluation indicated that 57% of Group I scored 9 to 11 points, and 43% scored 6 to 8 points. For Group II, 60% of the samples scored 9 to 11 points, 17% scored 6 to 8 points, and 23% scored 3 to 5 points. In Group III, 20% of the samples scored 9 to 11 points, 56% scored 6 to 8 points, and 24% scored 3 to 5 points. In Group IV, 34% of cases scored 9 to 11 points, 46% scored from 6 to 8 points, and 20% scored 3 to 5 points. For samples in Group V, 75% scored less than 3 points and 25% scored 3 to 5 points (Figure 8). The mean scores in Group I, II, III, IV, and V were 8.6, 8.1, 6.9, 7.3, and 1.6, respectively.

Table 3 shows the parameters calculated by adding up all the scores obtained by the rabbit groups.

There were statistical differences (*p* < 0.05) between Groups I and III and between Groups I and IV in the 52 weeks of observation (Table 4). Samples in Group V were excluded from the statistical analysis due to failure of the Mann–Whitney test criteria.

The distribution of results was similar in all observation periods (Table 5).

In Group I, 65% of the evaluated regenerates, and in Group II, 70% of the evaluated regenerates were filled with tissue with a 75–100% degree of similarity to the surrounding cartilage. In Groups III and IV, a similar amount of cavity-filling tissue was found in 39% and 57%, respectively, while in Group V, most cavities remained unfilled or were only 25% filled. Figure 9 shows images of the regenerates for Group II (Figure 9A—a cavity completely filled with tissue) and Group V (Figure 9B—a cavity slightly filled with cartilage), respectively.

### 4.5. Microscopic Evaluation

Each group of rabbits was considered in turn, and they were divided into subgroups on the basis of the number of points obtained according to the O’Driscoll scale (Table 1). Evaluation of the data indicated that 43% of samples in Group I scored 16 to 20 points, and 57% scored 12 to 15. In Group II, 54% of the samples scored 16 to 20 points, 17% scored 12 to 15 points, 16% scored 8 to 11 points, and 13% scored 0 to 7 points. In Group III, 28% of the samples scored 16 to 20 points, 44% scored 12 to 15 points, 26% scored 8 to 11 points, and 2% scored 0 to 7 points. In Group IV, 35% of cases scored 16 to 20 points, 27% scored 12 to 15 points, 31% scored 8 to 11 points, and 7% scored 0 to 7 points. For Group V, 92% of the samples scored less than 7 points and 8% scored 8 to 11 points (Figure 10). The mean scores in Group I, II, III, IV, and V were 15, 13.6, 13, 13, and 4, respectively.

In Group I rabbits, the average score was 15 (Table 6). There was a statistically significant improvement in the quality of regenerates during the longest observation period compared to the shortest. The best quality of regenerated tissues was observed after a one-year healing period, with 70% of results being very good. There was also a significant advantage in the results in Group I compared to those in Group IV after a one-year observation period (Table 7). In Group II rabbits, the average score was 13.6 (Table 6). There was no significant difference in the quality of the regenerates due to the length of the observation period (Table 8), nor were there significant differences between this and the other groups (Table 7). In Group III, the average score was 13 (Table 6). Extending the observation period did not significantly improve the quality of the regenerates (Table 8). As in Group II, there were no significant differences between IV and the other groups (Table 7). In Group IV, the average score was 13 (Table 6). The length of the observation period did not significantly affect the quality of the regenerates (Table 8). After a one-year observation period, the regenerate scores were significantly lower than that of Group I (Table 7). In Group V (the control group), the average score was 4 (Table 6), with 90% of the scores not exceeding 8 on the O’Driscoll scale (Figure 11A), and the length of the healing period did not significantly affect regeneration. In groups, I, II, III, and IV, tissue similar to mature hyaline cartilage predominated. In Group V, regenerates composed of fibrocartilage predominated (Figure 11D), accounting for 56%.

In Groups I and II, 70% were regenerated with normal cellularity (Figure 12A). In Group V, regenerates with necrotic features predominated (Figure 12B). There were no significant differences in the results among any of the described groups that were dependent on the length of the observation period.

Group I and II rabbits showed mostly complete regeneration of subchondral bone, with an increase in the number of normal results over longer periods of observation (Figure 13A). In Groups III and IV, subchondral bone regeneration occurred to a similar degree as in Groups I and II, but no improvement in regeneration scores was observed over longer observation periods. Group V was dominated by regenerates in which regeneration of the subchondral layer did not exceed 50% (Figure 13B).

## 5. Discussion

Various types of three-dimensional (3-D) matrices have been used for the regeneration of articular cartilage defects in research work carried out over the last fifty years. These include both natural and synthetic materials [57,58,59]. This is the first report describing the effectiveness of chondrocyte transplantation placed on a polyethersulfone (PES) membrane. In this study, a synthetic PES matrix was used for the first time to regenerate deep defects of the articular cartilage in rabbit knee. The properties of PES membranes were compared with commercially available collagen membranes used in the treatment of articular cartilage defects in humans.

The PES scaffold used in our research was produced based on collaboration between chemists and orthopedic surgeons. The desired features of the scaffold are:Sterile and non-cytotoxic;Preservable without losing its properties;Allows culturing of cells in a 3-D structure [60,61,62];Biodegradable after 6 months.

PES, which is a semipermeable membrane, possesses a pore system and a large inner surface that promotes cell migration and attachment. Due to its wide-meshed interconnecting pores, extensive contact between cells is possible. The porous, spongy structure of the membrane, with its interconnecting pores, promotes the development of collagen fibrils. The specific structure of the PES membrane and its durability (it is a non-absorbable polymer) protects the formation of vulnerable tissue, on one hand, while on the other hand it enables the complete disintegration of the scaffold after about six months, allowing undisturbed development of the cartilage. PES membranes contain sulfate groups [SO_2_] attached to the main chain of the polymer. These groups in the membranes mimic similar sulfate groups in the negatively charged glycosaminoglycan chains of the hyaline cartilage matrix. In this study, the PES membrane does not have a skin layer, which on one hand could be a significant obstacle to cell penetration, and cells would stick to the surface of the membrane and would not penetrate inside it. On the other hand, such an epidermal layer could protect joint-transplanted cells from escaping into the joint or from the impact of bone marrow-derived immune cells on the forming cartilage in allogeneic chondrocyte grafts.

The PES membrane has a higher resistance to tearing than collagen in an aqueous environment. This semipermeable material can be held in position by sutures and/or tissue glue, preventing membrane migration under mechanical loading.

The Chondro-Gide collagen membrane is a two-layer structure. The more compact layer—corresponding to the epidermal layer of the membrane—is a reinforcing layer, ensuring the strength of the carrier, and protecting the cells inhabiting the membrane from escaping into the synovial fluid, as well as against mechanical injuries. The less compact, spongy layer, made of collagen fibers with a loose structure, is inhabited by cells. The collagen structure stimulates chondrocytes cultivated in vitro to adopt cartilage metabolism and produce type II collagen and proteoglycans, according to the information provided by the manufacturer of the membrane. The arrangement of the collagen fibers increases the membrane’s resistance to tearing, allowing it to be attached with surgical sutures, and preventing it from detaching from the joint surface when the joint is subjected to loading. The Chondro-Gide collagen membrane covering the defect of the joint surface, attached to the margins of the defect with surgical sutures and sealed with tissue glue, can serve as a barrier for the suspension of tissue culture cells introduced under its surface. This method of collagen membrane implantation is currently being used in the treatment of defects of the human cartilage surface. Brittberg reported the treatment of full-thickness defects of cartilage in the knee with chondrocyte transplantation with monolayer culture. This method seems to be suitable for the treatment of cartilage defects with autologous chondrocytes, but there is a risk that cells transplanted in suspension may leak out of the defect due to joint motion [63].

In the described research, collagen and PES scaffolds were placed in six-well cell culture plates to enable their colonization by cells, cell proliferation within the spatial structure of these membranes, and the production of a cartilage matrix containing type II collagen and proteoglycans during culture. A similar method of cultivating cartilage cells on collagen scaffolds was described by Behrens et al. [64].

Our observations showed that the collagen membrane, in aqueous environments such as tissue culture, lost its biomechanical properties, taking on the consistency of a gel, preventing its transfer during culture and implantation into the joint. Compared to the PES scaffold, whose properties did not change under the influence of the environment, collagen seems to be less useful as a cell carrier.

Cartilage fragments taken from rabbit knee joints were exposed to a proteolytic enzyme to isolate the cells. Following Brittberg [65], collagenase and deoxyribonuclease were used for the enzymatic isolation of cartilage cells according to the current knowledge. Despite existing reports on the use of trypsin for the isolation of chondrocytes [66], due to its cytotoxic effect, described by Moskalewski [67], it was not used in the experiment. After 12 h of isolation, about 1.5–2 × 10^5^ cells were obtained from one rabbit, which is half the value reported by Brittberg [65]. After 4 days of culture, most of the cells were attached to the bottom of the culture plate or settled on the membranes. Cells attached to the bottom changed shape from spherical to spindle-like, similar to fibroblasts. At 14 to 21 days after the beginning of cultivation, as a result multiple divisions, a single layer of cells was formed at the bottom of the six-well cell culture plate. Similar observations have been made by other researchers [65].

A major problem with the regeneration of cartilage by autologous chondrocyte implantation is the dedifferentiation of chondrocytes during monolayer culture, resulting in cells with a fibroblast-like phenotype. These cells produce fibrous tissue instead of hyaline cartilage. Matmati et al. described the use of heat-inactivated allogeneic serum (HIFBS) in a monolayer expansion of bovine chondrocytes to generate cells with differentiated phenotypes [68]. In our study, the three-dimensional arrangement of the PES membrane under in vitro conditions enhanced cell proliferation, and the dedifferentiation of chondrocytes to a fibroblast phenotype was not observed.

According to Benya and Schaffer [69], the chondrocytes isolated from the matrix are phenotypically unstable and dedifferentiate under monolayer culture conditions. They take a form similar to that of mesenchymal progenitor cells or prechondrogenic precursor cells, which are similar to fibroblasts. The effect of their metabolism is the production of proteoglycans and type I and III collagen, which are not characteristic of cartilage [70]. The re-envelopment of the extracellular matrix restores the cells to their chondrocytic form. It is highly probable that the isolation and dedifferentiation of chondrocytes in vitro is a way to obtain a large number of mesenchymal cells that, when implanted into the joint, continue the production of the cartilage matrix started in culture for as long as they do not redifferentiate [71].

In the elementary analysis and after taking pictures of the membranes by means of electron microscopy, two weeks after the start of culture, the presence of tissue on the PES scaffold was observed. The available literature does not contain information on the cultivation of any cells on a PES membrane; therefore, the results described in this study can be regarded as innovative. The average tissue mass of about 0.23 mg on each fragment of membrane with a diameter of 0.5 cm, as well as the electron microscopy images of both the surface and the inside of the membrane after cell culture, prove that the sulfone polymer is a material that provides adequate conditions for the newly formed tissue. It enables undisturbed growth of cells in the 3-D environment, which may reduce the potential for chondrocytes to assume a form similar to fibroblasts, which is characteristic of cultures in a single layer at the bottom of the six-well plate.

The collagen structure of the Chondro-Gide membrane and the nitrogen content made it difficult to perform an elementary analysis of this membrane. Its protein chemical composition makes it impossible to objectively analyze the nitrogen content, on the basis of which the amount of protein on the PES scaffold was estimated. No cells were isolated from the collagen membrane in order to estimate their number, despite the attempts made. The cells isolated and propagated in vitro expressing the gene encoding type II B collagen, which is a molecular marker of cartilage tissue [72,73], were characterized by a metabolism typical for chondrocytes. The expression of the type I collagen gene, specific for fibroblasts, proves that the cells were dedifferentiated in culture and took on a form similar to prechondrogenic fibroblast-like mesenchymal cells [69,70]. According to some authors, the cultivation of cartilage cells at the bottom of the culture flask causes their irreversible dedifferentiation, and stops the production of type II collagen and proteoglycans [74].

Culturing chondrocytes on a porous PES membrane creates spatial conditions for proliferating cells that are completely different from those that are possible when cultured in a single layer. It can be assumed that cartilage cells undergo less dedifferentiation under the conditions created by the porous sulfone polymer. The significantly lower intensity of the bands at the level of 400 base pairs, i.e., the lower expression of the type I collagen gene for cells collected from the PES scaffold in comparison to the cells from the bottom of the wells in the culture plate, may indicate the preserved metabolism characteristic of cartilage tissue.

Summarizing the macroscopic evaluation, the advantage in terms of the points obtained by Groups I and II over III and IV is visible. The average results and the frequency of very good results were both higher in these groups. There is a subtle advantage, although not statistically confirmed, of the scores obtained in the group in which cells were transplanted on a collagen membrane compared to the group where PES was used as a carrier. The statistical analysis did not confirm a significant advantage in terms of the points obtained for Group I over II; however, a significant difference was shown between Group I rabbits and both groups when using only membranes—PES and collagen—in the longest observation period of fifty-two weeks. There were also no significant differences between the quality of regenerates after transplantation of empty membranes, apart from a very slight advantage to the results obtained for Group IV—the PES scaffold—compared to III—the collagen membrane. The results obtained for the control group (V), not included in the statistical analysis, were significantly worse than those obtained for the groups after cell implantation on membranes and empty membranes.

In the microscopic assessment of cartilage surface repair, analysis of the sum of the scores obtained in individual groups indicated a slight advantage in the results obtained for Group I after the transplantation of cartilage cells placed on collagen material compared to Groups II, III and IV, in which the described results were similar to each other. There were no statistical differences between these groups: only in Group I was there a statistically significant improvement in the results after the longest period of observation. In the group left to heal spontaneously (V), the regeneration of damage in the cartilaginous surface had not occurred even after the longest observation period. The differences in results between Groups I, II, III and IV were less pronounced than those reported by Brittberg et al. [65]. He noted and statistically confirmed the advantage of the quality of regenerates in the group of animals in which chondrocytes were transplanted compared the group receiving periosteum transplantation. The difference described may be related to the poorer regenerative potential of the periosteum compared to collagen and PES scaffolds.

Pulkkinen et al. [75] tested a recombinant human type II collagen (rhCII) gel combined with autologous chondrocytes as a scaffold for cartilage repair in rabbits in vivo. Similar to our studies, cultivation for 2 weeks prior to transplantation into a lesion with a diameter of 4 mm created in the rabbit’s femoral trochlea. After 6 months, the repair tissue in both groups filled the lesion, but with rhCII repair, the filling was more complete. The O’Driscoll grading showed no significant differences between rhCII repair and spontaneous repair, with both presenting lower quality than intact cartilage. When rhCII was used to repair cartilage defects, the repair quality was histologically incomplete, but still, the rhCII repair showed moderate mechanical characteristics, which were a slight improvement over those following spontaneous repair.

Allogenic chondrocytes may be rejected by the defect as a result of the immune response. Articular cartilage is thought to have low immunogenicity, because the cellular antigens of chondrocytes are covered by the extracellular matrix. However, once chondrocytes have been isolated from the extracellular matrix, they do show immunogenicity [76]. The chondrocytes used in our experimental model were autologous and immunogenic rejection could not be induced.

To overcome the disadvantages of autologous osteochondral transplantation in the treatment of deep osteochondral defects, Schleicher I. et al. developed two biphasic scaffolds—a hydroxyapatite/collagen scaffold and an allogenous sterilized bone/collagen scaffold—and tested their integration in a sheep model. A moderate lowering of the surface, smaller defect size, fewer immune-competent cells in the specimens, and significant upregulation of collagen II and SOX-9 messenger ribonucleic acid expression on the surface of the allogenous sterilized bone/collagen scaffold compared with the hydroxyapatite/collagen scaffold were observed [77]. Recent tissue-engineering approaches, including gene delivery, have been proposed for the regeneration of cartilage tissue [78]. Odabas S. and Co. investigated genetically modified cells with plasmid-encoding bone morphogenetic protein-7 (BMP-7) implanted into gelatin-oxidized dextran scaffolds in the regeneration of auricular cartilage defects in New Zealand (NZ) white rabbits.

## 6. Conclusions

The established method for the isolation and culturing of chondrocytes is adequate to provide a sufficient number of cells that can be used as a transplant. Autologous chondrocyte transplantation resulted in the growth of the regenerated tissue, revealing a morphology similar to hyaline-like cartilage. The quality of the regenerated tissue after the transplantation of the PES membrane was similar to that for the collagen membrane. The RT-PCR analysis of mRNA isolated from the chondrocytes cultured on the PES scaffolds showed the expression of type II procollagen. Elemental analysis of PES membrane slices after 2 weeks of culture with chondrocytes showed a concentration of 0.23 mg of tissue per membrane section. Macroscopic and microscopic evaluation indicated that the quality of the regenerated tissue was similar after the transplantation of cells placed on PES or collagen membranes. The regenerated tissue in the full-thickness defects reached maturity after 12 weeks, and exhibited a hyaline cartilage morphology even after 52 weeks.

## Figures and Tables

**Figure 1 polymers-15-02360-f001:**
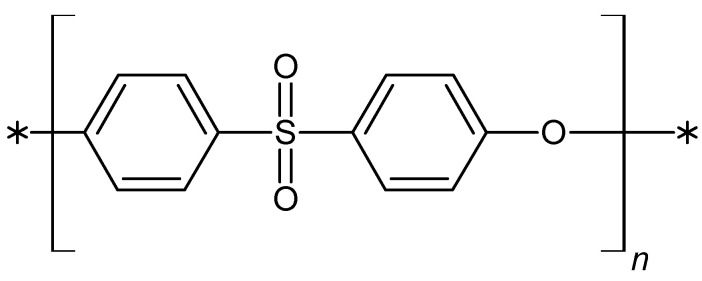
The chemical formula of the polyethersulfone, where *n* = 50–80.

**Figure 2 polymers-15-02360-f002:**
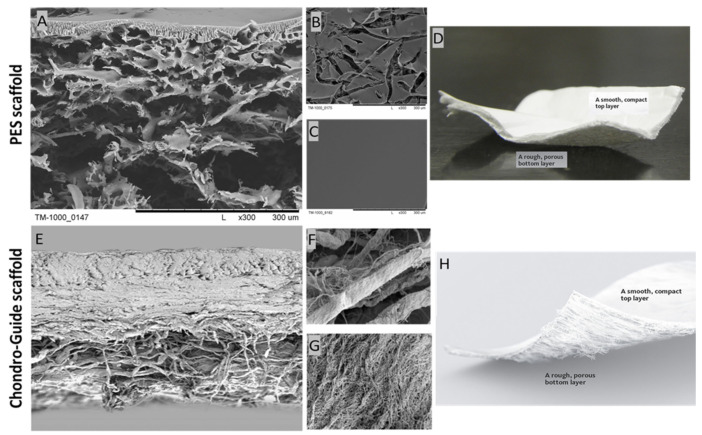
SEM photomicrographs of the PES scaffold: (**A**) cross-section; (**B**) bottom layer; (**C**) top layer; and the Chondro-Gide scaffold: (**E**) cross-section; (**F**) top layer; (**G**) bottom layer. Photos of the PES scaffold (**D**) and the Chondro-Gide scaffold (**H**). Adapted from [51], Geistlich. Magnification: (**A**–**C**) ×300; (**E**) ×100; (**F**,**G**) ×1500.

**Figure 3 polymers-15-02360-f003:**
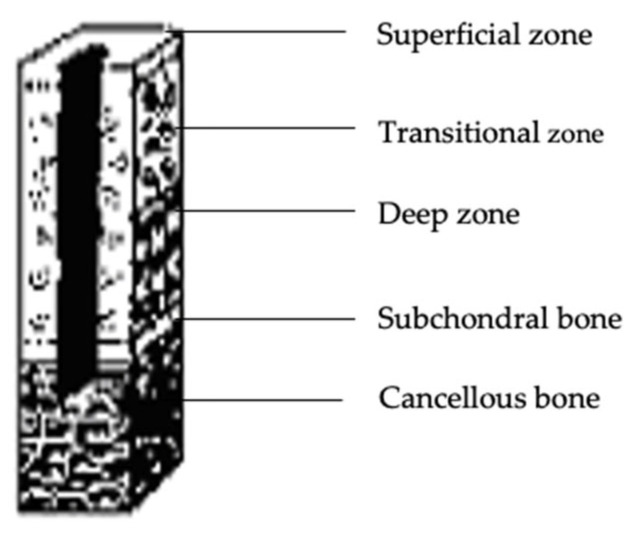
Articular cartilage loss exceeding the subchondral bone—grade IV cartilage damage according to the ICRS scale Adapted from [52], InTech, 2011.

**Figure 4 polymers-15-02360-f004:**
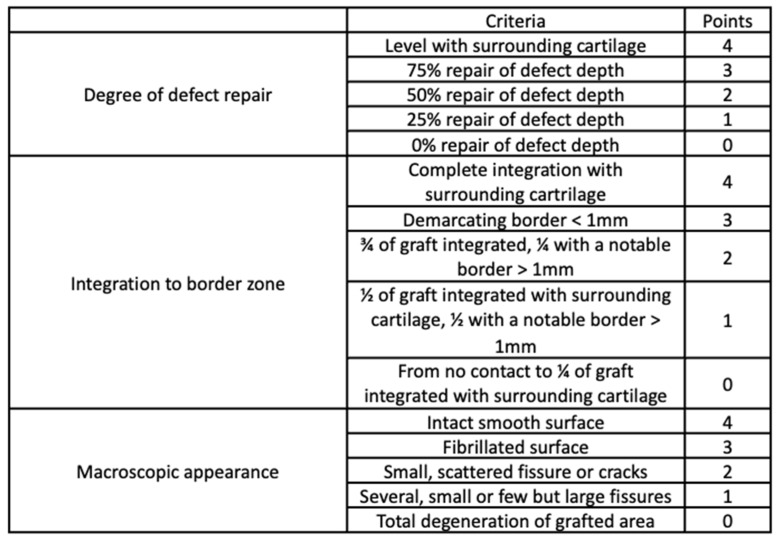
International Cartilage Repair Society (ICRS) cartilage repair assessment tool. Adapted from [54], Oxford University Press, 2019.

**Figure 5 polymers-15-02360-f005:**
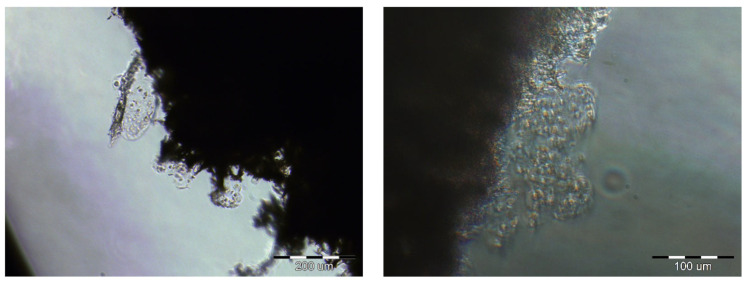
The cells on the edges of the PES scaffolds after 2 weeks of cultivation. Scale bars: left image—200 µm; right image—100 µm.

**Figure 6 polymers-15-02360-f006:**
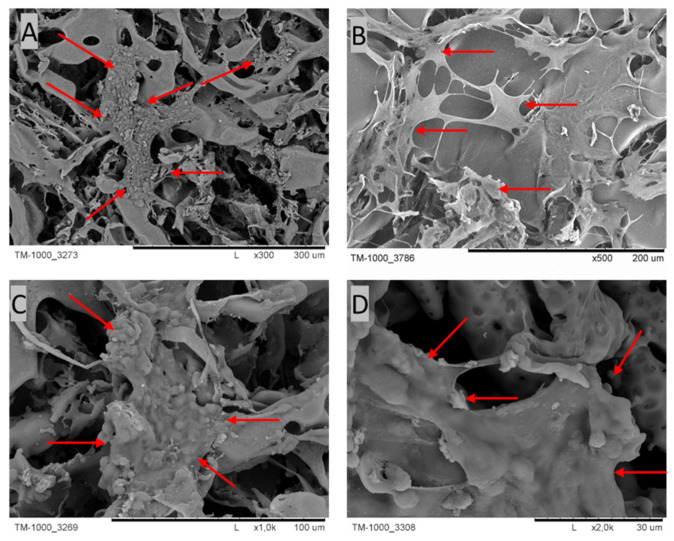
SEM micrographs of cross-sections and the top layers of PES scaffolds after 4 weeks of culture. The red arrows indicate cells with their ECM. Scale bars: (**A**) 300 µm; (**B**) 200 µm; (**C**) 100 µm; (**D**) 30 µm.

**Figure 7 polymers-15-02360-f007:**
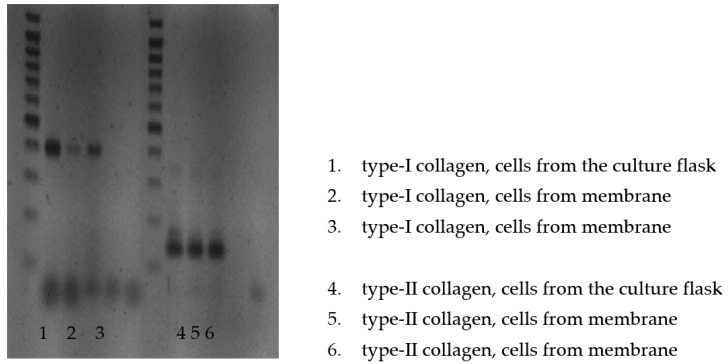
RT-PCR analysis of the mRNA isolated from the cells detached from the bottom of the culture plate and PES membrane.

**Figure 8 polymers-15-02360-f008:**
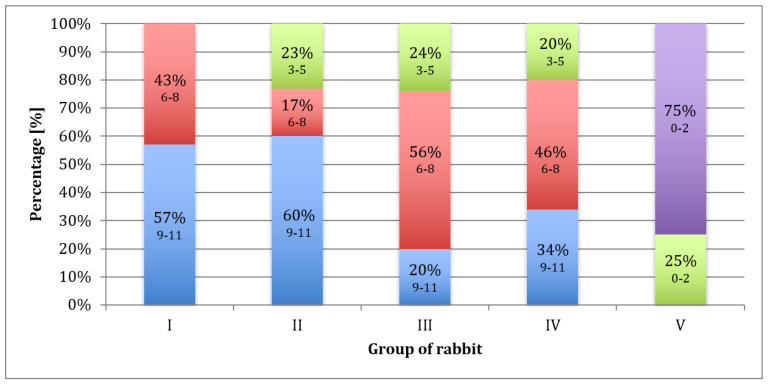
Percentage of rabbits that scored 9–11 (blue), 6–8 (red), 3–5 (green), and less than 3 (purple) points upon macroscopic evaluation after 52 weeks, according to the ICRS scale.

**Figure 9 polymers-15-02360-f009:**
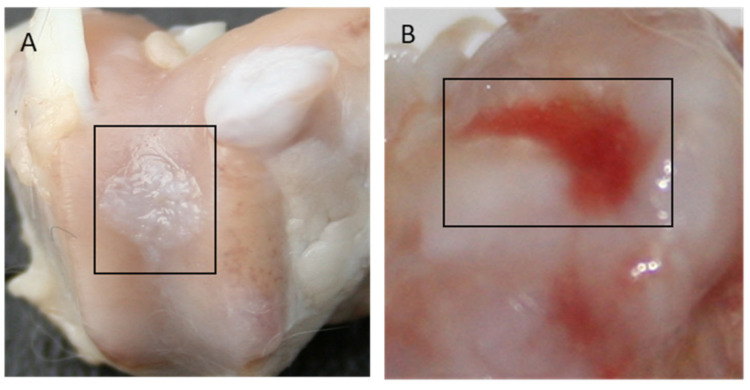
Gross appearance of the cartilage defect at 52 weeks after implantation for Group II (**A**) and Group V (**B**).

**Figure 10 polymers-15-02360-f010:**
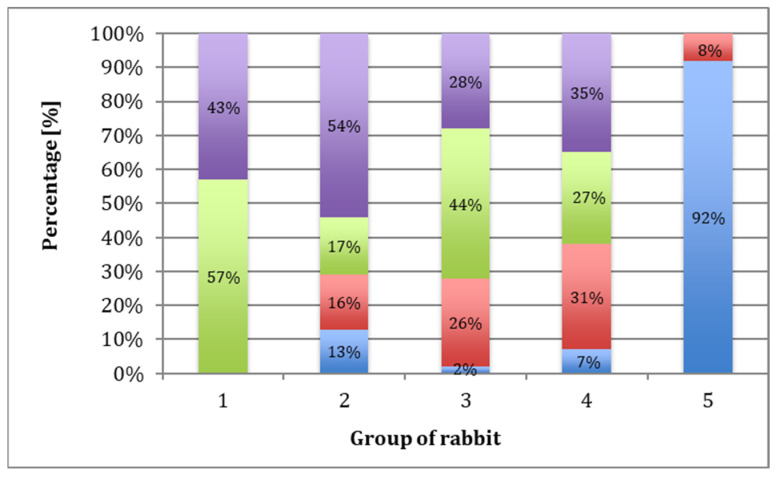
Percentage of rabbits that scored 16–20 (purple), 12–15 (green), 8–11 (red), and less than 7 (blue) points based on microscopic evaluation after 52 weeks, according to the O’Driscoll scale.

**Figure 11 polymers-15-02360-f011:**
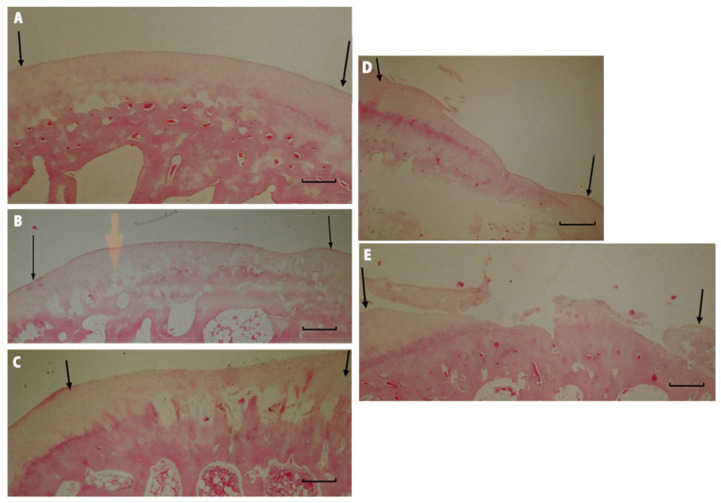
Microscopic imagines of histological samples of the cartilage of the studied groups. (**A**) Group I after 52 weeks of observation. The defect is bonded at both ends of graft (marked with arrows), the surface is smooth and intact, the thickness of the regenerated tissue is 100% that of normal cartilage, 50–99% of the subchondral bone is reconstructed. (**B**) Group II after 52 weeks of observation. The defect is well integrated, the surface is smooth, the thickness is 50–99% of normal cartilage, and more than 50% of the subchondral bone is reconstructed. (**C**) Group III after 52 weeks of observation. The surface of the regenerated tissue is smooth, the cartilage is well integrated, the thickness is 50–99% of normal cartilage, but less than 50% of subchondral bone is reconstructed. (**D**) Group IV after 52 weeks of observation. The surface of the defect is irregular, the margins of the defect are well integrated, the thickness is 50% of normal cartilage, and 100% of the subchondral bone is reconstructed. (**E**) Group V after 52 weeks of observation. The defect has disintegrated. Magnification x50.

**Figure 12 polymers-15-02360-f012:**
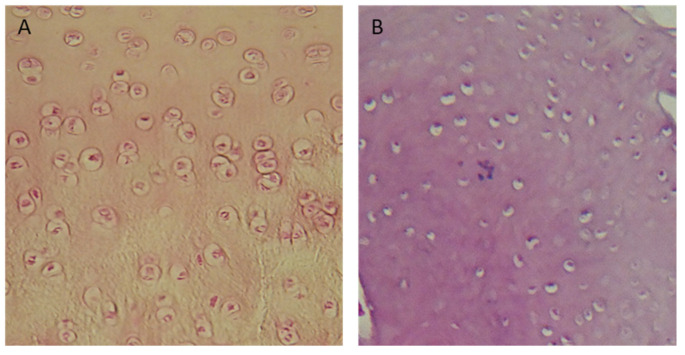
Microscopic images: (**A**) cell-rich regenerate, visible isogenic groups of cells (Group I); (**B**) Significant necrosis of regenerate, visible empty cavities. Magnification: (**A**) ×100; (**B**) ×200.

**Figure 13 polymers-15-02360-f013:**
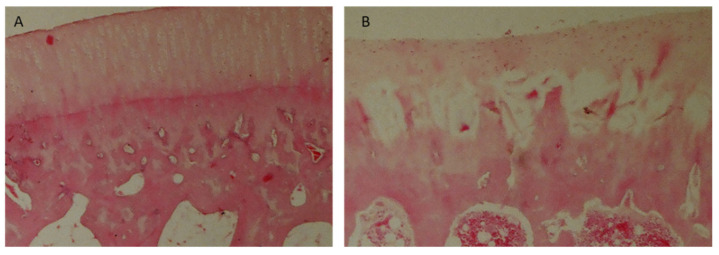
Microscopic images: (**A**) cell-rich regenerate, visible isogenic groups of cells (Group I); (**B**) significant necrosis of regenerate, visible empty cavities (Group III). Magnification: (**A**) ×100; (**B**) ×200.

**Table 1 polymers-15-02360-t001:** Criteria and scale for evaluating microscopic images of articular cartilage regeneration according to O’Driscoll [55,56].

Category	Subcategory	Characteristic	Score
1. Nature of predominant tissue	Cellular morphology	Hyaline articular cartilage	4
		Young hyaline cartilage	3
		Incompletely differentiated mesenchyme	2
		Fibrous cartilage	1
		Fibrous tissue or bone	0
2. Structural characteristics	Surface regularity	Smooth and intact	3
		Superficial horizontal lamination	2
		Fissures 25–100% of the thickness	1
		Severe disruption including fibrillation	0
	Structural integrity	Normal	2
		Slight disruption including cysts	1
		Severe disintegration	0
	Thickness	100% of normal adjacent cartilage	2
		50–99% of normal cartilage	1
		0–50% of normal cartilage	0
	Bonding to the adjacent cartilage	Bonded at both ends of graft	2
		Bonded at one end or partially at both ends	1
		Not bonded	0
3. Freedom from cellular changes of degeneration	Hypocellularity	Normal cellularity	3
		Slight hypocellularity	2
		Moderate hypocellularity	1
		Severe hypocellularity	0
	Degenerative changes	None	2
		Moderate	1
		Significant changes	0
4. Subchondral bone reconstruction		100%	2
		50–99%	1
		<50%	0

**Table 2 polymers-15-02360-t002:** Tissue mass content from PES scaffolds after 14 days of culture.

Number of PES Scaffold	Tissue Mass Content [mg]
1	0.28
2	0.19
3	0.25
4	0.21

**Table 3 polymers-15-02360-t003:** The parameters calculated by adding up all the grades achieved by the groups of rabbits.

	12 Weeks					25 Weeks					52 Weeks				
Groups	I	II	III	IV	V	I	II	III	IV	V	I	II	III	IV	V
mean	8.3	8.6	6.8	8	1.7	8.2	7.3	7	7.2	1.8	9.4	8.5	7	6.8	1.5
SD	1.5	2.8	1.3	2.4	1	1.7	2.6	1.3	2	1.3	1.5	2.5	2.1	2.1	1.1
median value	8	10	7	9	2	8	9	7	7	2	9	8.5	7	7	1.5

**Table 4 polymers-15-02360-t004:** Comparison of rabbits from Groups I, II, III, and IV: sum ratings based on the Mann–Whitney test.

Groups	Level of Statistical Significance of the Differences between Groups. Highlighted Fields Indicate Statistically Significant Differences (*p* ≤ 0.05)		
	12 Weeks	25 Weeks	52 Weeks
I/II	0.369	0.462	0.473
I/III	0.083	0.178	0.030
I/IV	0.930	0.354	0.029
II/III	0.083	0.534	0.178
II/IV	0.514	0.806	0.131
III/IV	0.312	0.962	0.962

**Table 5 polymers-15-02360-t005:** Results of the test to determine significant differences between the observation times (Mann–Whitney test).

Groups	Level of Statistical Significance of the Difference between Observation Times Highlighted Fields Indicate Statistically Significant Differences (*p* ≤ 0.05)		
	12 Weeks ↔ 25 Weeks	12 Weeks ↔ 52 Weeks	25 Weeks ↔ 52 Weeks
I	0.895	0.153	0.178
II	0.174	0.910	0.385
III	0.834	0.962	0.885
IV	0.596	0.470	0.700

**Table 6 polymers-15-02360-t006:** The parameters were calculated by summing the grades achieved by the rabbit groups.

	12 Weeks					25 Weeks					52 Weeks				
Groups	I	II	III	IV	V	I	II	III	IV	V	I	II	III	IV	V
mean	14	14	13	14	4	15	13	13	12	4	16	14	13	13	4
SD	1.5	6	3	4.4	1.2	1.7	4.7	2.9	3.4	2.2	2.1	4.3	4.1	3.2	1.1
median value	14	16	13.5	15	3	15	14.5	14.5	12	3	16.5	14.5	13	13.5	4.5

**Table 7 polymers-15-02360-t007:** Comparison between rabbit Groups I, II, III, and IV: sum ratings using the Mann–Whitney test.

Groups	The Level of Statistical Significance of Differences Between Groups. Highlighted Fields Indicate Statistically Significant Differences (*p* ≤ 0.05)		
	12 Weeks	25 Weeks	52 Weeks
I/II	0.221	0.462	0.326
I/III	0.441	0.441	0.066
I/IV	0.895	0.064	0.046
II/III	0.198	0.756	0.391
II/IV	0.596	0.744	0.424
III/IV	0.564	0.501	0.885

**Table 8 polymers-15-02360-t008:** The results of the test to determine the significant differences between the observation times (Mann–Whitney test).

Groups	Level of Statistical Significance of the Difference between the Observation Times Highlighted Fields Indicate Statistically Significant Differences (*p* ≤ 0.05)		
	12 Weeks ↔ 25 Weeks	12 Weeks ↔ 52 Weeks	25 Weeks ↔ 52 Weeks
I	0.233	0.018	0.121
II	0.307	0.650	0.406
III	0.431	0.847	0.847
IV	0.536	0.847	0.700

## Data Availability

Not applicable.

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
