# Peer review of "Intraarticular Implantation of Autologous Chondrocytes Placed on Collagen or Polyethersulfone Scaffolds: An Experimental Study in Rabbits"

_polymers, 2023, doi:10.3390/polym15102360_

Round 1

Reviewer 1 Report

The authors compared the cartilage regeneration ability between the synthesized polyethersulfone (PES) membrane and commercially available collagen membrane through animal experiments. The experimental groups and the control group were well established, and the reliability of the experimental results is increased by using a large number of rabbits. If a few deficiencies are corrected, it is expected to be published in the ‘Polymers’.

1.     At figure 2, the bottom layer of the PES membrane is too smooth. It is understood to have a smooth surface due to the effect of the wet inversion phase, but a detailed explanation should be added in the text to help readers understand.

2.     The authors divided the animals into five groups and conducted animal experiments. However, there is a significant difference in the number of animals used for each experimental group. The reason for the difference in the number of experimental animals should be clearly explained.

3.     For the tissue mass content in line 291, the standard deviation should be stated along with the mean value.

4.     It should be indicated in the graph how many weeks figure 10 and figure 12 are the results of the experiment.

5.     In case of Figure 13 and 14, for all experimental conditions (I to V), the results of the same implant site should be shown, and 5 types of pictures should be supplied as one picture set. In addition, enlarged images of characteristic results (hyaline cartilage production) should be provided.

Author Response

Dear Reviewer 1,

Thank you for your comments, suggestions, tips. I appreciate the time and effort that you have dedicated to providing your valuable feedback on my manuscript. Below I put your comments along with my response.

Point 1:

At figure 2, the bottom layer of the PES membrane is too smooth. It is understood to have a smooth surface due to the effect of the wet inversion phase, but a detailed explanation should be added in the text to help readers understand.

Response 1:

Thank you for your comment. Taking this into account, I added detailed explanation in the manuscript (in materials section).

Point 2:

The authors divided the animals into five groups and conducted animal experiments. However, there is a significant difference in the number of animals used for each experimental group. The reason for the difference in the number of experimental animals should be clearly explained

Response 2:

Thank you for your comment. In this research, we performed the largest number of trials in groups I and II due to the implantation of membranes with cells into the cartilage defects after a two-week culture. In groups III and IV, we implanted only empty membranes. Group V, the smallest control group, in which the results significantly differed from groups I and II as well as III and IV, was not included in the statistical analysis. In order to compare the mentioned groups, the results obtained in each group were compared in the statistical analysis using the non-parametric Mann-Whitney test. The level of statistical significance of the difference between the results obtained in groups I, II, III, IV and the level of statistical significance of the difference between the results obtained in particular observation times were calculated.  P < 0.05 was assumed as a statistically significant difference.

Point 3:

For the tissue mass content in line 291, the standard deviation should be stated along with the mean value.

Response 3:

Thank you for your comment. Taking this into account, it has been corrected.

Point 4:

It should be indicated in the graph how many weeks figure 10 and figure 12 are the results of the experiment.

Response 4:

Thank you for your comment. I added time in the description of figure 10 and 12.

Point 5:

In case of Figure 13 and 14, for all experimental conditions (I to V), the results of the same implant site should be shown, and 5 types of pictures should be supplied as one picture set. In addition, enlarged images of characteristic results (hyaline cartilage production) should be provided.

Response 5:

Thank you for your comment. It has been corrected.

Yours sincerely,

Maciej Płończak

Reviewer 2 Report

This study compared the PES membrane with the collagen membrane for cartilage defect repair in a rabbit model. The following comments are provided for the authors’ consideration:

1.     The abstract should include the main findings and conclusion, which are missing in the abstract of this manuscript. “Hyaline cartilage has poor repair capabilities, leading to progressive joint damage.” The sentence is not correct.

2.     The introduction didn’t highlight the novelty and necessity of this study. There has been lots of paper reporting the usage of PES or collagen for cartilage defect repair. In addition, the discussion is unfocused and didn’t highlight the importance of the findings in this study compared to previous studies.

3.     The materials and methods are suggested to be integrated as one part as there are some overlapped descriptions.  The sequence of the methods could be adjusted, the elemental analysis and SEM observation could be placed after PCR, which corresponds to their sequence in the result.

4.     Figure 1, there is no “n” in the figure. In the legend, “n=50 ÷ 80”, please correct.

5.     The figure 2, 3, 4 are suggested to be arranged in one figure to compare the membrane structures directly and should be placed at the result part.

6.     Figure 5 is unclear to illustrate the structure of the osteochondral unit.

7.     Line 171, where are the “non-weight-bearing area of the articular surface” for cell isolation? It should be clearly pointed out.

8.     The depth of the defect was 4mm, the membrane shown in figure 3 is quite thin, how the authors make sure the defect was properly filled with the membrane and how the membrane was fixed at the lesion site.

9.     Why in SEM observation and PCR, the collagen membrane results are absent since its collagen composition would not affect the results in these two methods.

10.  The PCR result should be quantified.

11.  For macroscopic and histological analysis, the images for each group should be provided.

12.  The quality of histological figures must be improved. Histological images should show the margins of the defects; bars should be added in the images.

13.  Except for HE staining, other staining methods such as Safarin O, or Alcian blue are useful to illustrate the matrix deposition, which should be used here.

14.  Since PES is not degradable, did authors observe the PES on the histological images?

15.  Line 33, “Articular cartilage (AC), which covers the distal parts of the bones”, “distal part” is not accurate.

16.  Line 52-55, punctuation is missing.

17.  Line 60, “formation of the regenerate”, please correct.

18.  Line 90, the synthetic polymers lack cell binding sites, so “provide cell attachment” is not an advantage of synthetic polymers compared to nature polymers.

19.  Some sentences should be re-written in a scientific way, for example, Line 101, “Non-filled cavities were used as a control”. Line 40, “This whole thing can even lead to disability”.

  The overall writing should be improved in a scientific way.

Author Response

Dear Reviewer 2,

Thank you for your comments, suggestions, tips. I appreciate the time and effort that you have dedicated to providing your valuable feedback on my manuscript. Below I put your comments along with my response.

Point 1: The abstract should include the main findings and conclusion, which are missing in the abstract of this manuscript. “Hyaline cartilage has poor repair capabilities, leading to progressive joint damage.” The sentence is not correct.

Response 1:

Thank you for your comment. Taking this into account the abstract has been changed.

Point 2:  The introduction didn’t highlight the novelty and necessity of this study. There has been lots of paper reporting the usage of PES or collagen for cartilage defect repair. In addition, the discussion is unfocused and didn’t highlight the importance of the findings in this study compared to previous studies.

Response 2:

Thank you for your comment. It has been corrected.

Point 3: The materials and methods are suggested to be integrated as one part as there are some overlapped descriptions.  The sequence of the methods could be adjusted, the elemental analysis and SEM observation could be placed after PCR, which corresponds to their sequence in the result.

Response 3:

Thank you for your comment. It has been corrected

Point 4: Figure 1, there is no “n” in the figure. In the legend, “n=50 ÷ 80”, please correct.

Response 4:

Thank you for your comment. It has been corrected.

Point 5: The figure 2, 3, 4 are suggested to be arranged in one figure to compare the membrane structures directly and should be placed at the result part.

Response 5:

Thank you for your comment. The figure 2, 3, 4 have been arranged in one figure

Point 6: Figure 5 is unclear to illustrate the structure of the osteochondral unit.

Response 6: Thank you for your comment. In all the knee joints the grade IV defects penetrating into subchondral bone were performed. We can remove the figure 5.

Point 7: Line 171, where are the “non-weight-bearing area of the articular surface” for cell isolation? It should be clearly pointed out.

Response 7:

Thank you for your comment. It has been changed.

Point 8: The depth of the defect was 4mm, the membrane shown in figure 3 is quite thin, how the authors make sure the defect was properly filled with the membrane and how the membrane was fixed at the lesion site.

Response 8: Thank you for your comment. The thickness of PES scaffolds was about 350–400 µm. Sometimes when the thickness was not sufficient, the membranes were double-rolled. It should be taken into account that these were post-culture membranes, with cells.

Point 9: Why in SEM observation and PCR, the collagen membrane results are absent since its collagen composition would not affect the results in these two methods.

Response 9: Thank you for your comment. In this work, the comparison of both membranes was made in terms of macroscopic and microscopic evaluation. Due to the collagen structure of the membrane, no other tests were performed.

Point 10: The PCR result should be quantified.

Response 10: Thank you for your comment.

No quantified results were done in this study.

Point 11: For macroscopic and histological analysis, the images for each group should be provided.

Response 11: Thank you for your comment. We have added pictures of histological figures for each group. Due to the significant similarity of macroscopic images in groups I, II, III, IV, we did not include images for individual groups apart from the control group V.

Point 12: The quality of histological figures must be improved. Histological images should show the margins of the defects; bars should be added in the images.

Response 12: Thank you for your comment. We have improved the quality of histological figures, histological images show the margins of the defects, we added bars in the images.

Point 13: Except for HE staining, other staining methods such as Safarin O, or Alcian blue are useful to illustrate the matrix deposition, which should be used here.

Response 13: Thank you for your comment. We don’t have any other staining methods except for HE staining.

Point 14: Since PES is not degradable, did authors observe the PES on the histological images?

Response 14: Thank you for your comment. Polysulfone is not absorbed but breaks down into monomers. In the performed microscopic observations, polysulfone monomers were not observed in any of the four groups of rabbits, regardless of the length of healing time.

Point 15: Line 33, “Articular cartilage (AC), which covers the distal parts of the bones”, “distal part” is not accurate.

Response 15: Thank you for your comment. Articular cartilage (AC), which covers the articular surfaces. It has been corrected.

Point 16: Line 52-55, punctuation is missing.

Response 16: Thank you for your comment. It has been corrected.

Point 17: Line 60, “formation of the regenerate”, please correct.

Response 17: Thank you for your comment. It has been corrected.

Point 18: Line 90, the synthetic polymers lack cell binding sites, so “provide cell attachment” is not an advantage o

f synthetic polymers compared to nature polymers.

Response 18: Thank you for your comment. It has been corrected.

Point 19: Some sentences should be re-written in a scientific way, for example, Line 101, “Non-filled cavities were used as a control”. Line 40, “This whole thing can even lead to disability”.

Response 19: Thank you for your comment. It has been corrected.

Yours sincerely,

Maciej Płończak

Reviewer 3 Report

Poor repair ability of hyaline cartilage, leading to degeneration of articular cartilage. In this manuscript, the authors proposed the Autologous Chondrocyte Implantation (ACI). on two different scaffolds for the treatment of lesions in hyaline cartilage in rabbits. First one was collagen type I/III commercial scaffold 19 (Chondro–Gide) and the second one was a synthetic polyethersulfone (PES) membrane. This author cultured rabbit derived chondrocytes onto the scaffold and implanted them into the cartilage defect area of the rabbit to repair the cartilage defect. This is research with great practical application value article. Thus, it merits to be published. I just suggest some major modifications before publication:

(1) Please introduce the main results of this study in the Abstract in concise language. If necessary, please rewrite Abstract part.

(2) The use of biomaterials should first consider the issue of cytotoxicity. Please supplement the cytotoxicity experimental data using PES scaffold materials.

(3) Please explain the reason for this result “Collagen type I/III contain nitrogen so the results would not be correct” in page 10 line 289-290.

(4) Many biological materials show biocompatible on the body. Please refer to the literature (Journal of Materials Science & Technology, 2022, Doi: 10.1016/j.jmst.2022.07.008) to supplement the reasons for PES scaffold materials in “Introduction” part.

(5) Please supplement the primer sequence used for DNA amplification by PCR.

(6) Most recent studies about biological materials were evaluated, such as Bioactive Materials 18 (2022) (Doi: 10.1016/j.bioactmat.2022.01.048); Adv. Funct. Mater. 2021 (Doi: 10.1002/adfm.202009432). Biomater Transl. 2020. (DOI: 10.3877/cma.j.issn.2096-112X.2020.01.002). Biomater Transl. 2022.   (DOI: 10.12336/biomatertransl.2022.02.001), etc., are recommended to be cited in proper places.  

This manuscript still requires English polishing to avoid colloquial expression.

Author Response

Dear Reviewer 3,

Thank you for your comments, suggestions, tips. I appreciate the time and effort that you have dedicated to providing your valuable feedback on my manuscript. Below I put your comments along with my response.

Point 1: Please introduce the main results of this study in the Abstract in concise language. If necessary, please rewrite Abstract part.

Response 1: Thank you for your comment. Taking this into account the abstract has been changed.

Point 2: The use of biomaterials should first consider the issue of cytotoxicity. Please supplement the cytotoxicity experimental data using PES scaffold materials.

Response 2: Thank you for your comment. In the Materials Section – “Membranes” we supplemented some studies which show that PES membrane was used with cell studies.

Point 3: Please explain the reason for this result “Collagen type I/III contain nitrogen so the results would not be correct” in page 10 line 289-290.

Response 3: Thank you for your comment. Collagen membranes were not included in the elementary analyzes due to the nitrogen content in type I and III collagen contained in these membranes. In elementary analysis, we measure the nitrogen content in protein of tissue origin. In the case of a collagen membrane, the nitrogen content would distort this measurement. Another reason was also a change of the consistency of the collagen membrane after two weeks of cell culture, which took the form of a dissolved gel.

Point 4: Many biological materials show biocompatible on the body. Please refer to the literature (Journal of Materials Science & Technology, 2022, Doi: 10.1016/j.jmst.2022.07.008) to supplement the reasons for PES scaffold materials in “Introduction” part.

Response 4: Thank you for your comment. I has been refered.

Point 5: Please supplement the primer sequence used for DNA amplification by PCR.

Response 5: Thank you for your comment. The primer sequence has been added.

Point 6: Most recent studies about biological materials were evaluated, such as Bioactive Materials 18 (2022) (Doi: 10.1016/j.bioactmat.2022.01.048); Adv. Funct. Mater. 2021 (Doi: 10.1002/adfm.202009432). Biomater Transl. 2020. (DOI: 10.3877/cma.j.issn.2096-112X.2020.01.002). Biomater Transl. 2022.   (DOI: 10.12336/biomatertransl.2022.02.001), etc., are recommended to be cited in proper places.  

Response 6: Thank you for your comment. The articles have been cited.

Yours sincerely,

Maciej Płończak

Round 2

Reviewer 2 Report

Some issues must be addressed before this manuscript could be accepted for publication: 

1.     The abstract is too long, please make it brief and highlight the main findings and conclusion. The description “Scaffolds were made of collagen type I/III and polyethersulfone (PES).” is confusing because actually the authors is saying that two kinds of scaffolds were compared, one scaffold is made of collagen, the other is made of PES. In addition, the revised abstract is still lack of conclusion.

2.     The discussion is unfocused and didn’t highlight the importance of the findings in this study compared to previous studies.

3.     the discussion is unfocused and didn’t highlight the importance of the findings in this study compared to previous studies.

4.     The primer sequences should be placed in the method not in the result.

5.     Figure 1, there is no “n” in the figure. In the legend, “n=50 ÷ 80”, please correct.

6.     In Figure 5, the authors are suggested to mark the name of each layer in the image to illustrate the defect penetrating into the subchondral bone layer.

7.     In the animal experiment, how the membrane was fixed at the lesion site without detachment. Please add in the method of the animal experiment.

Author Response

Dear Reviewer 2,

Thank you for your comments, suggestions, tips. I appreciate the time and effort that you have dedicated to providing your valuable feedback on my manuscript. Below I put your comments along with my response:

  1. The abstract has been corrected.

2, 3. In the discussion, the properties of both types of membranes were discussed and compared. The advantages of the PES membrane in cell culture in three-dimensional conditions are described. The results of cartilage formation on both membranes were analyzed in comparison with other authors using other types of scaffolds.

  1. The primer sequence has been placed in the methods.
  2. Figure 1 has been changed.
  3. It has been corrected.
  4. It has been added.

Yours sincerely,

Maciej Płończak

Round 3

Reviewer 2 Report

All my comments are addressed.

Author Response

Thank you for your positive review
